# Indwelling Peritoneal Catheter for Ascites Management in a UK District General Hospital: A Cohort Study

**DOI:** 10.3390/healthcare9101254

**Published:** 2021-09-24

**Authors:** Karl Jackson, Katie Frew, Robert Johnston, Joanna Coleman, Leonie Armstrong, Avinash Aujayeb

**Affiliations:** 1Respiratory Department, Northumbria Healthcare NHS Foundation Trust, Cramlington NE23 6NZ, UK; karl.jackson@nhct.nhs.uk; 2Palliative Care Department, Northumbria Healthcare NHS Foundation Trust, Cramlington NE23 6NZ, UK; katie.frew@nhct.nhs.uk (K.F.); joanna.coleman@nhct.nhs.uk (J.C.); leonie.armstrong@nhct.nhs.uk (L.A.); 3Acute Medicine Department, Northumbria Healthcare NHS Foundation Trust, Cramlington NE23 6NZ, UK; robert.johnston@nhct.nhs.uk; 4Northumbria HealthCare NHS Foundation Trust, Care of Tracy Groom, Northumbria Way, Cramlington NE23 6NZ, UK

**Keywords:** ascites, malignant ascites, indwelling peritoneal catheter

## Abstract

Background: There is no national or local guidance for management of malignancy-related ascites (MRA). Modalities can include large volume paracentesis (LVP) and indwelling peritoneal catheter (IPeC) insertion. Objectives: We set up a local IPeC service and performed a retrospective review with local ethical (Caldicott) approval. We hypothesized that an IPeC service would reduce inpatient stay related to MRA management, would be acceptable to patients, and have minimal complications. Methods: Notes of all patients requiring IPeC insertion were reviewed. Descriptive statistical methodology was applied with continuous data presented as mean (standard deviation (SD); range) and categorical variables as frequencies or percentages. Integrated Palliative Care Outcome Scale (IPOS) scores were collected for IPeC patients. Results: Thirty-four patients were identified. They were predominantly female, with a mean age of 66.6 years and a wide range of cancer diagnoses. Twenty-nine were inserted as day case procedures, and 31 had preceding paracenteses (mean 2). Main complications were leakage (6(17%)), peritonitis (2(5.8%)), and skin infection (1(3%)). IPOS scores showed consistent improvement in symptoms. Conclusions: An IPeC service for malignant-related ascites is acceptable to patients and is associated with manageable complication rates. We present the development of our service and hope for widespread application.

## 1. Introduction

Ascites describes the accumulation of fluid in the peritoneal cavity. It is associated with liver cirrhosis, congestive cardiac failure, hypoalbuminemia, renal failure, and malignancy. Malignant-related ascites (MRA) is a more appropriate term than “malignant ascites,” as ascites development in malignancy does not imply peritoneal dissemination. Peritoneal carcinomatosis can cause lymphatic blockage and increase vascular permeability. Hepatic metastases can cause MRA by fluid production by tumour cells, compressive or obstructive portal hypertension, or liver failure [1,2].

The appearance, cytological, microbiological, and biochemical analyses help in the determination of the aetiology. If the serum-ascites albumin gradient (SAAG) (subtracting the ascitic value from the serum value) is ≥ to 1.1 g per decilitre (g/dL), portal hypertension is present [1,2].

Median survival with MRA (excluding ovarian cancer) is 4 to 16 weeks. MRA causes significant morbidity and reduced quality of life. Symptoms include abdominal distension, negative body image, orthopnoea, dyspnoea, nausea, vomiting, constipation, loss of appetite, and pain [1,2].

There is no national guidance for MRA management. Modalities include large volume paracentesis (LVP), salt restriction and diuretics if portal hypertension, peritoneal-venous shunts (PVS), trans-jugular intrahepatic portosystemic shunts (TIPS), and long-term indwelling peritoneal catheters (IPeCs) [3,4]. PVS and TIPS are commoner in non-malignant refractory ascites. Antibiotic prophylaxis against bacterial peritonitis is also recommended if portal hypertension is present [3,4]. The aim of this study is to review the service development and describe patient outcomes.

## 2. Materials and Methods

Local Caldicott approval was obtained for a service evaluation of all patients who underwent IPeC placement in NHCT, from October 2018 to May 2021 (reference 7439). Demographics and outcomes were collected. Patients were identified prospectively and sequentially but analysed all at once, retrospectively. Descriptive statistical methodology was applied with continuous data presented as mean (standard deviation (SD) or range) and categorical variables as frequencies or percentages. Integrated Palliative Care Outcome Scale (IPOS) scores were collected for the IPeC cohort on the day of insertion. Patients were supplied with the forms to fill in at 7 and 14 days and asked to post back. We used the STROBE case-control checklist when writing our report [5].

Care in Northumbria Healthcare NHS Foundation Trust (NHCT) is organized across four main hospitals: three “base sites” and one acute care centre. The latter was a flagship hospital, has 210 beds, was part of NHS England’s New Models of Care programme, and has a purpose-built wing for medical ambulatory care (MAC). Patients with MRA are initially seen in MAC by the acute care medicine team. An IPeC service was developed, and we offer patients with MRA the option of having an IPeC at the time of their first drainage if and when MRA re-accumulates [6].

All IPeCs are placed in theatre or dedicated clean spaces by a gastrointestinal surgeon, an interventional respiratory consultant, a pleural fellow, or an acute medicine staff grade. Rocket^®^ Peritoneal IPC™ systems are used locally. Pre-operative antibiotics are administered. The specialist palliative care team coordinate referrals, community drainage regimens, and overall care. All are performed as day case procedures.

### IPC Technique

The patients are placed in the supine position, and an abdominal point of care ultrasound is performed to mark the spot for incision and insertion of the IPeC. The marked area is sterilized and draped. Local anaesthetic is then administered at the marked site. Two incisions are made about 5 cm apart. A tract is created with straight forceps and the drain passed through, making sure that the cuff of the drain sits midway. A dilator and a sheath are passed through the proximal incision into the peritoneal cavity. The drain is threaded through the sheath into the peritoneal space, and the sheath peels off. A vacuum bottle is connected to drain, or the drain can be connected to a catheter bag to allow large-volume drainage.

## 3. Results

Thirty-four patients (17 male, 17 female) underwent 35 IPeC placements (one patient had two IPeCs). Diagnoses were gastrointestinal (GI) (50% (17)), ovarian (17% (5)), breast (7.4% (2)), prostate (3% (1)), adrenal (3% (1)), thymic (3% (1)), unknown primary (14.7% (6)) cancers, and mesothelioma (3.7% (1)). Fourteen patients did not have ascitic fluid cytology sent, but all had diffuse peritoneal carcinomatosis on imaging. Of the 20 samples that were sent, 12 (60%) were positive (gastric—2, pancreatic—2, ovarian—4, unknown primary—2, breast—1, colorectal—1). Six were negative (hepatocellular cancer—1, pancreatic—1, ovarian—1, adrenal cancer—1, colorectal—1, gastric—1, prostate—1). The World Health Organization Performance Status (WHO PS) was 1 in 7 patients, 2 in 8 patients, 3 in 15 patients, and 4 in 4 patients (2—symptomatic, <50% in bed during the day, 3—symptomatic, >50% in bed but not bedbound, 4—bedbound).

Three patients had no preceding LVP; 31 (91%) had a mean of two LVPs before. Six (22.22%) developed post-operative leaks due to fluid bypassing around the tube. These were managed with additional suturing and dressings. Two (5.8%) patients developed cellulitis; swabs in both grew fully sensitive staphylococcus aureus. Bacterial peritonitis occurred in one (3%) patient (concurrent growth of pseudomonas oryzihabitans and staphylococcus epidermidis) and was successfully treated. Four IPeCs were removed: two as MRA resolved, one for tumour infiltration, and one for non-resolving site cellulitis. Twenty-nine patients died; median number of days to death was 37 (range 6–262). IPOS scores, collected in 20 patients, consistently show reduction in symptoms. Figure 1 summarises IPOS scores pre and post IPeC insertion at the time of discharge after the ascites was drained. Forms given to the patients to be returned at 7 and 14 days were incomplete and not analysable.

No patients required further procedures for ascites drainage.

## 4. Discussion

This study is the first description of a IPeC service in the United Kingdom for malignant ascites. The local IPeC service is safe and effective [6]. The patients’ IPeCs were inserted as day procedures, and subsequent ascites drainage could be managed in an ambulatory setting, making it an excellent option to palliate symptoms related to MRA. NICE recommends vacuum-assisted drainage of treatment-resistant, recurrent malignant ascites with vacuum drainage systems [4] Estimated cost saving is approximately £1051 per patient. Formal health economic analysis is beyond the scope of this article.

Point of care ultrasound, pre-operative antibiotics, and performing the procedure in a dedicated clean space by experienced practitioners enables risk reduction. We also advocate large-volume drainage at the time of IPeC insertion so that leaks due to the pressure of large volume ascites are reduced. This is purely local expert opinion. District nurses, who perform the majority of the drainages, are certified in the aseptic technique required.

The patient described as having bacterial peritonitis above had two organisms growing in his ascitic fluid (taken from the IPeC), which showed 50% of polymorphs and 50% of lymphocytes at the time of analysis. It was unclear whether this was bacterial colonisation or actual peritonitis, but high inflammatory markers settled on antibiotics. The sample was taken from the catheter rather than a new ascitic tap: bacterial colonisation in IPeCs is currently being debated in the literature [7,8], although studies are required in a greater selection of patients.

The IPOS scale is well validated in the palliative care setting. The scale incorporates a series of wide-ranging questions that elicit patient reported concerns. It is also brief and can be used in those with advanced illness. They have also been used in assessing quality of life in IPeCs in non-malignant ascites [9]. The IPOS sheets for the IPeC patients were not completed fully or always returned, and thus, we do not know if the improvement was sustained. However, there is no study looking at the minimal clinically important difference for IPOS scores. This is part of the limitations of our study.

Our study also has no control arm. However, we believe that sharing our initial findings and protocol will enable other care systems to benefit by applying our system to their practice. We would welcome further collaboration with any centres.

Furthermore, not everyone had positive cytology, but as Karoo et al. found, only approximately 60% of ascitic fluid cytology is positive, and thus other factors, such as the presence of peritoneal carcinomatosis, come into play. Our positive cytology rate is exactly 60%. Since our study was performed, a protocol (Appendix A) was devised mandating sending for cytology as well as biochemical tests on any MRA [10]. Regular service reviews are planned over the next five years. We have also not collected data on costs and overall length of stay related to MRA.

There are many unanswered questions: the optimal time to insert an IPeC and whether it should be offered as a first line intervention, whether improvement of quality of life is sustained over time, how quickly does bacterial colonisation occur after insertion, and whether this equates to dormant peritonitis or a non-clinically significant issue. There is also no clear guidance on management of malignant-related ascites and no national registry of patients having the procedures to elucidate the who, what, when, where, and associated outcomes.

We have applied for funding with the proposal to perform an open-labelled prospective feasibility study where patients with MRA either have a paracentesis and then an indwelling catheter if they so choose or an indwelling peritoneal catheter at their first presentation. We propose to collate outcomes electronically on quality of life using the IPOS scores over 7, 14, 30, and 60 days as well as any complications arising from the catheters. It would be important to develop the minimal clinically important difference for IPOS scores in patients with malignant ascites. We propose analysing samples of peritoneal fluid at the above times to assess for the presence of bacterial colonisation and whether this affects outcomes. We predict that the attrition rate will be high at 30 days but are hopeful that over time, we would recruit a suitable cohort. We also propose to canvass national societies and send out surveys to determine who places these indwelling catheters and whether an indwelling peritoneal catheter registry could be set up.

## 5. Conclusions

IPeC placement for malignant-related ascites is beneficial to patients and should be implemented where resources allow service development. Future implications for research are optimising the timing of IPeC placement, determining whether bacterial colonisation is a clinically relevant problem and whether reduction in IPOS scores is sustainable over time.

## Figures and Tables

**Figure 1 healthcare-09-01254-f001:**
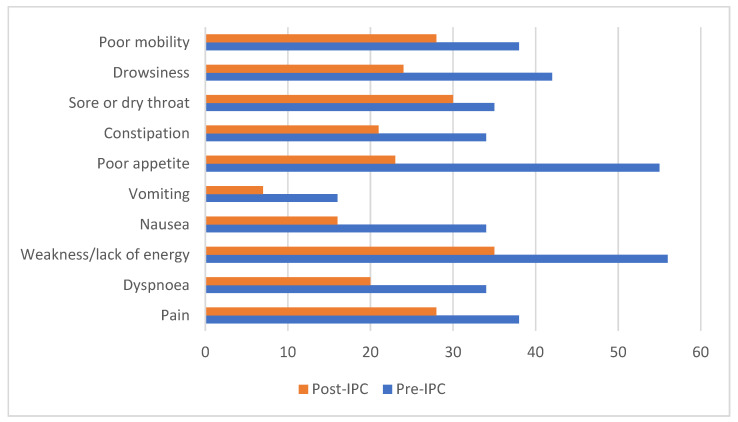
Showing reduction in IPOS score per symptom, pre and post IPeC insertion.

## Data Availability

AA has full access to the data and can supply that if a reasonable request is made.

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
