# Peer review of "Indwelling Peritoneal Catheter for Ascites Management in a UK District General Hospital: A Cohort Study"

_healthcare, 2021, doi:10.3390/healthcare9101254_

Round 1

Reviewer 1 Report

Greetings to the authors on this study on IPC.

This is a good observational data on the use of indwelling peritoneal catheter for the use of malignancy related ascites. 

Draw back of the study:

  1. Observational data
  2. Small sample size.

Will be interesting to see what the IPOS scores will be at 60 days on these patients.

This is a good observational study and I recommend this for publication.

Author Response

Thank you- there are no specific points required for this. We have already mentioned the 2 points below in our limitations and thus, we thank the reviewer for these kind comments. Avinash 

Reviewer 2 Report

Thank you for the opportunity to review this manuscript by K Jackson and colleagues titled:

Indwelling peritoneal catheter for ascites management in a UK district general hospital: a cohort study.

This manuscript is a descriptive study on individuals with malignant-related ascites who underwent an indwelling peritoneal catheter is an important topic where data is sparse.

Overall, this study represents an important addition to the literature in managing individuals with malignancy-related ascites. I believe this manuscript can be better by addressing these few comments.

Introduction

The focus here should be more relevant to indwelling peritoneal catheter service development. 

An aim should be specified.

Paragraph 5, the last line seems out of place? a more detailed care pathway should be described or reference to supp material. Are all patients with MRA offered an IPC or is it on recurrence? Are the IPeC inserted as a day procedure.

I feel paragraphs five, six, and section 1.1 should be added in the methodology section

Methods

Is this a retrospective study? How were the records obtained? Did the service maintain a prospective database

The IPOS score should be described further to make sense of the results. Is there a minimal clinical important difference? HAs it been used in indiviudal with malignancy related ascites?

Did all patients have malignancy confirmed on cytology?

Was functional status obtained?

Does post-operative leak represent ascitic fluid bypassing around the tube?

Were other important patient related outcome measures captured? i.e. intiail hospital length of stay, further peritoneal procedures or hospital admissions related to malignancy related ascites?

Survival - why use mean as the range describe shows that the data is not normally distributed.

Discussion

This is a bbit confusing as it is hard to follow and not addressing the findings of the study.

There is also additional informaton about cost and patients who did not have a IPeC

Minor comments

Preferably the abbreaviation IPC should be replaced with IPeC to make it less confusing to the readers

Author Response

Thank you for the opportunity to review this manuscript by K Jackson and colleagues titled:

Indwelling peritoneal catheter for ascites management in a UK district general hospital: a cohort study.

This manuscript is a descriptive study on individuals with malignant-related ascites who underwent an indwelling peritoneal catheter is an important topic where data is sparse.

Overall, this study represents an important addition to the literature in managing individuals with malignancy-related ascites. I believe this manuscript can be better by addressing these few comments.

Introduction

The focus here should be more relevant to indwelling peritoneal catheter service development. 

An aim should be specified.

we have added some remarks to that effect

Paragraph 5, the last line seems out of place? a more detailed care pathway should be described or reference to supp material. Are all patients with MRA offered an IPC or is it on recurrence? Are the IPeC inserted as a day procedure.

we have added our local protocol which has been developed, and yes we offer IPCs after recurrence. All IPCs are inserted as day case

I feel paragraphs five, six, and section 1.1 should be added in the methodology section

we agree and have moved the sections

Methods

Is this a retrospective study? How were the records obtained? Did the service maintain a prospective database

we maintained the patient numbers prospectively, but all the data was analysed retrospectively

The IPOS score should be described further to make sense of the results. Is there a minimal clinical important difference? HAs it been used in indiviudal with malignancy related ascites?

this is an important point. there is no such studies. we have noted this in our limitation. we might seek to develop this, but this is a major piece of work and perhaps beyond the scope of our group. if the reviewer wanted to reach out to us, we could collaborate on this. the IPOS has not been used in MRA before, but used in non-malignant ascites (reference 11). 

Did all patients have malignancy confirmed on cytology?

This is an important point, and we have elaborated on this. Not everyone had positive cytology, but those who didnt had peritoenal carcinomatosis. Rates of positive cytology approach 50% ( Karoo ROSLloyd TDRGarcea G, et al How valuable is ascitic cytology in the detection and management of malignancy?

Was functional status obtained? yes we did in terms of the performance status

Does post-operative leak represent ascitic fluid bypassing around the tube? yes it did, this is mentioned, and how me mitigate agaisnt that by fully draining at the time of insertion

Were other important patient related outcome measures captured? i.e. intiail hospital length of stay, further peritoneal procedures or hospital admissions related to malignancy related ascites? - no, this is part of the limitations and mentioned but no further drainage procedures were required

Survival - why use mean as the range describe shows that the data is not normally distributed.- we have now used median

Discussion

This is a bbit confusing as it is hard to follow and not addressing the findings of the study.

I think we did address the findings of the study- we described the feasibility of the local set up, and the fact that it is effective, and the IPOS scores suggest relief of symptoms of patients. However, we do agree that it might be hard to follow, and thus we have re-arranged some of the sections and expanded on the limitations. 

There is also additional informaton about cost and patients who did not have a IPeC

this is an error and has been removed

Minor comments

Preferably the abbreaviation IPC should be replaced with IPeC to make it less confusing to the readers done

Round 2

Reviewer 2 Report

Thank you.

Minor comments:

Introduction: I would add aims at the end of the introduction.

Methods: No further comments

Results: LAst line of the second paragraph - I would remove sustained as only pre- and post-procedure IPOS scores were obtained.

I am unsure is post-procedure IPOS scores represent immediately after procedure or at day 30 etc.

Discussion

Paragraph one needs to be expanded a bit more on the findings and I would remove the cost difference between pleurX and Rocket

E.g.

"This study report the local Northumbrian IPeC experience (is this the first UK study?? is so can change it to the first UK experience etc.) of management of malignancy-related ascites. Overall, this study shows that a dedicated IPeC service was safe and effective. The patient's IPeC was inserted as a day procedure and subsequent ascites drainage could be managed in an ambulatory setting making it an excellent option to palliate symptoms related to MRA".

It will be useful to add a bit more information regarding IPOS to paragraph 2. Some of the information regarding IPOS can be moved here.

Author Response

Introduction: I would add aims at the end of the introduction.

This has been done. 

Methods: No further comments

Results: LAst line of the second paragraph - I would remove sustained as only pre- and post-procedure IPOS scores were obtained.

this has been removed

I am unsure is post-procedure IPOS scores represent immediately after procedure or at day 30 etc.

It is not immediate, but some hours afterwards when the ascites has been fully drained from the Ipec. I have made this clearer

Discussion

Paragraph one needs to be expanded a bit more on the findings and I would remove the cost difference between pleurX and Rocket

E.g.

"This study report the local Northumbrian IPeC experience (is this the first UK study?? yes is so can change it to the first UK experience etc.) of management of malignancy-related ascites. Overall, this study shows that a dedicated IPeC service was safe and effective. The patient's IPeC was inserted as a day procedure and subsequent ascites drainage could be managed in an ambulatory setting making it an excellent option to palliate symptoms related to MRA- this is an excellent sentence and perhaps we can borrow it in its entirety? ".

It will be useful to add a bit more information regarding IPOS to paragraph 2. Some of the information regarding IPOS can be moved here.

I have re-arranged some of the paragrahps and added more details. i think it is better now. thank you